# Epigenetic Age in Prader–Willi Syndrome and Essential Obesity: A Comparison with Chronological and Vascular Ages

**DOI:** 10.3390/jcm14051470

**Published:** 2025-02-22

**Authors:** Antonello E. Rigamonti, Valentina Bollati, Benedetta Albetti, Diana Caroli, Adele Bondesan, Graziano Grugni, Silvano G. Cella, Alessandro Sartorio

**Affiliations:** 1Department of Clinical Sciences and Community Health, University of Milan, Dipartimento di Eccellenza 2023-2027, 20129 Milan, Italy; silvano.cella@unimi.it; 2EPIGET Lab, Department of Clinical Sciences and Community Health, University of Milan, Dipartimento di Eccellenza 2023-2027, 20122 Milan, Italy; valentina.bollati@unimi.it (V.B.); benedetta.albetti@unimi.it (B.A.); 3Occupational Health Unit, Fondazione IRCCS Ca’ Granda Ospedale Maggiore Policlinico, 20122 Milan, Italy; 4Experimental Laboratory for Auxo-Endocrinological Research, Istituto Auxologico Italiano, IRCCS, 28824 Piancavallo-Verbania, Italy; d.caroli@auxologico.it (D.C.); a.bondesan@auxologico.it (A.B.); g.grugni@auxologico.it (G.G.); sartorio@auxologico.it (A.S.)

**Keywords:** epigenetic age, chronological age, vascular age, Prader–Willi syndrome, essential obesity

## Abstract

**Background**: Prader–Willi syndrome (PWS) is a rare genetic disorder mapping to the imprinted 15q11-13 locus, specifically at the paternally expressed snord116 region, which has been implicated in controlling epigenetic mechanisms. Some aspects of the PWS-related clinical phenotype, such as the high mortality rate in adulthood, might be attributed to accelerated epigenetic ageing. **Objectives**: The aim of the present case–control study was to evaluate epigenetic age, age acceleration, vascular age (VA), and vascular ageing in adults with PWS (*n* = 24; F/M = 11/13; age = 36.8 [26.6; 45.3] years; body mass index, BMI = 36.8 [33.9; 44.8] kg/m^2^), compared with a sex- and age-matched group of subjects with essential obesity (EOB) (*n* = 36; F/M = 19/17; age = 43.4 [30.6; 49.5] years; BMI = 44.8 [41.2; 51.7] kg/m^2^). **Results**: In subjects with PWS, there was a younger epigenetic age and a lower age acceleration than in subjects with EOB. No differences were found between VA and vascular ageing in the two groups. Epigenetic age was associated with chronological age and VA within each group. For each group, no relevant associations of epigenetic age or age acceleration with demographic, biochemical, and clinical parameters were found. When considering individuals with PWS, there were no associations of epigenetic age with growth hormone (GH) deficiency, duration of hormone replacement therapy, and plasma levels of insulin-like growth factor 1 (IGF-1). **Conclusions**: The hypothesis of accelerated epigenetic ageing in PWS should be rejected. Additionally, considering the existence of a SNORD116-dependent epigenetic dysregulation in PWS, the results of the present study might be misleading, since an epigenetics-based approach was used to measure ageing.

## 1. Introduction

Prader–Willi syndrome (PWS) is a so-called rare disease, genetically determined, and characterised by a variable clinical phenotype, including muscle hypotonia, failure to thrive during the neonatal period, short stature, mental disability, hyperphagia, and (even severe) obesity in adolescence and adulthood [1,2,3].

The genetic basis of PWS involves imprinted genes on the proximal long arm of chromosome 15. This results in the absence of function of genes normally expressed in a monoallelic fashion only from the paternal chromosome. In 60–70% of individuals with PWS, the genetic defect is a deletion in the area of 15q11-13 on the paternal chromosome 15 (del15), while a further 25–30% present with uniparental disomy (UPD15) for maternal chromosome 15. Paternal del15 and maternal UPD15 are functionally equivalent, as they both result in the absence of a paternal contribution to the genome in the 15q11-13 region. The small nucleolar RNAs C/D box (SNORDs) has been identified to have a critical role in the 15q11-13 region, being putatively part of the imprinting centre that regulates the expression of several genes in the transcriptional domain, among which there is the snord116 gene. Two rare causes of PWS are imprinting mutations, which are microdeletions or point mutations, and translocations with specific breakpoints, all involving the PWS gene region [4,5,6].

Among several endocrinopathies, growth hormone (GH) deficiency is frequently observed in individuals with PWS. When diagnosed in childhood, this condition is promptly treated with hormone replacement therapy based on recombinant human GH (rhGH), which may continue into adulthood in approximately 20% of cases [7,8].

Some clinical studies in adults with PWS, mainly those non-treated (or inadequately treated?) with rhGH, show an increased risk of age-related diseases in an earlier phase of their lives (e.g., diabetes mellitus [DM], cardiovascular diseases [CVDs], and cognitive decline) [9,10]. The mortality rate of individuals with PWS has been estimated to be 3% per year up to adulthood and 7% per year after 30 years old [11].

One possible reason for these epidemiological observations might be an acceleration of epigenetic age, as occurs in progeria and other similar syndromic conditions [12,13,14,15].

Epigenetics studies the biological–molecular mechanisms underlying the regulation of gene expression, cell function, and phenotype, which do not depend on changes in the DNA sequence. Epigenetic modifications, particularly those deriving from DNA methylation, which, continuously and even reversibly, are inserted in the chromatin with advancing age in every living being, have been reported to track epigenetic ageing. There is still doubt about whether these drive the molecular and cellular processes of ageing, such as age-related inflammation, oxidative stress, and mitochondrial dysfunction (an active driver of epigenetic ageing), or, alternatively, represent an epiphenomenon, being the consequence of molecular and cellular damage accumulating over time (passive by-stander of epigenetic ageing). In both cases, epigenetic modifications define chronological and epigenetic ages, which, in physiological conditions, should coincide [16].

In this regard, the concept of an “epigenetic clock” has been proposed as a valuable tool for calculating an individual’s epigenetic age [17].

Methodologically, the epigenetic clock is based on the measurement of DNA methylation levels in CpG sites of specific gene loci (which are different depending on the algorithm used); for each individual, it is possible to determine their epigenetic age, which can be compared with the corresponding chronological age, permitting the calculation of the so-called age acceleration, i.e., the difference between the epigenetic and chronological ages [18,19].

Interestingly, there is strong evidence that epigenetic age is increased in obesity [20], and weight loss for diet and/or exercise has been reported to reduce epigenetic age, particularly in subjects with a positive age acceleration, an effect presumably due to the improvement of the chronic inflammatory state [21].

Obesity is also characterised by an increased CVD risk, due to the association with several CVD-related risk factors: DM, atherosclerosis, dyslipidaemia, hypertension, heart failure, myocardial infarction, etc. Apart from smoking habit, these considerations also explain why vascular age (VA), an alternative measure representing an individual’s CVD risk in terms of years, is typically increased in obesity [22,23]. Interestingly, VA has been reported to positively correlate with biological age, evaluated through different methodological approaches such as epigenetic age [21].

According to this view, similar to essential obesity (EOB), in PWS, there is a shortening of telomers, which are short, repeated nucleotide sequences found at the end of linear chromatids to protect the genetic information, a biological–molecular event that marks cellular senescence. Interestingly, neither hormone replacement therapy based on rhGH nor CVD risk factors have been capable of explaining this (apparently) accelerated ageing in individuals with PWS [24].

To the best of our knowledge, no one has evaluated epigenetic age in individuals with PWS apart from the shortening of telomeres. Compared to VA, epigenetic age might permit demonstrating the existence of age acceleration in individuals with PWS and foreseeing the impact of epigenetic ageing and CVD risk factors on mortality in this clinical condition.

Thus, the present study aimed to determine epigenetic ageing and VA in adults with PWS, compared to a sex- and age-matched group of subjects with EOB. We hypothesise that in individuals with PWS there is accelerated epigenetic ageing, which parallels increased VA, a possible cause of the high mortality rate in adulthood.

## 2. Materials and Methods

### 2.1. Study Design and Subjects

The present case–control study was prospective.

Adults of both sexes were selected from the patient population admitted to the Division of Auxology (for individuals with PWS) and the Division of Metabolic Diseases (for individuals with EOB) of Istituto Auxologico Italiano, Piancavallo-Verbania, Italy, for a 3-week in-hospital multidisciplinary body weight reduction program.

First, individuals with PWS were selected; then, age- and sex-matched individuals with EOB were identified and included in the control group.

Individuals with PWS showed the typical clinical phenotype of the syndrome, with this diagnosis being confirmed by cytogenetic analysis (del15 = 20 cases; UPD15 = 4 cases).

GH deficiency was present in 10/24 subjects with PWS (at the moment of blood sampling for the present study), while all these cases were treated with rhGH; the duration of rhGH treatment was 16.8 [12.0; 19,2] years when it was continued into adulthood, and 7.0 [2.0; 9.0] years when it was completed in paediatric age. The plasma level of IGF-1 in all PWS patients, including those without GH deficiency, was 150.0 [89.5; 184.5] ng/mL.

The inclusion criteria for the individuals of both groups (i.e., PWS and EOB) were as follows: (1) both genders, (2) age ≥ 18 years, and (3) body mass index (BMI) > 30 kg/m^2^. The exclusion criteria were as follows: (1) the presence of secondary causes of obesity (e.g., steroid-induced obesity), apart from PWS; (2) systolic blood pressure (SBP) ≥ 180 mmHg and diastolic blood pressure (DBP) ≥ 110 mmHg; (3) cardiovascular, psychiatric, neurological, or other (relevant) medical diseases evident in the previous 6 months; and (4) refusal to sign the informed consent form by the patients (and/or by the parents of subjects with PWS, when necessary).

Some of these patients were taking medications for diabetes (metformin and/or insulin), hypertension, and dyslipidaemia (see Results for details).

The study protocol was approved by the Ethical Committee (EC) of the Istituto Auxologico Italiano, IRCCS, Milan, Italy (EC code: 2022_03_15_04; research project code: 01C213; acronym: ETABIOLPWS).

### 2.2. Anthropometric Measurements

Height and weight were measured using a scale with a stadiometer (Wunder Sa.Bi., WU150, Trezzo sull’Adda, Italy). Waist circumference (WC) was measured with a flexible tape in a standing position, halfway between the ribs’ inferior margin and the crista’s superior border. Body composition was determined by bioimpedance analysis (Human-IM Scan, DS-Medigroup, Milan, Italy) after the subject was at supine rest for 20 min. BMI, fat mass (FM), and fat-free mass (FFM) were determined in all subjects.

### 2.3. Metabolic Variables

Blood samples (about 10 mL) were collected at around 8:00 AM after an overnight fast.

Total cholesterol (T-C), high-density lipoprotein cholesterol (HDL-C), low-density lipoprotein cholesterol (LDL-C), triglycerides (TG), glucose, insulin, and high-sensitivity C-reactive protein (hsCRP) were measured.

Serum T-C, LDL-C, HDL-C, and TG levels were determined using colorimetric enzymatic assays (Roche Diagnostics, Monza, Italy). The sensitivities of the assays were 3.86 mg/dL [1 mg/dL = 0.03 mmol/L], 3.87 mg/dL [1 mg/dL = 0.03 mmol/L], 3.09 mg/dL [1 mg/dL = 0.03 mmol/L], and 8.85 mg/dL [1 mg/dL = 0.01 mmol/L], respectively.

Serum glucose level was measured using the glucose oxidase enzymatic method (Roche Diagnostics, Monza, Italy). The method’s sensitivity was 2 mg/dL [1 mg/dL = 0.06 mmol/L].

Serum insulin level was determined by a chemiluminescent immunometric assay using a commercial kit (Elecsys Insulin, Roche Diagnostics, Monza, Italy). The method’s sensitivity was 0.2 µIU/mL [1 µU/mL = 7.18 pmol/L].

hs-CRP was measured using an immunoturbidimetric assay (CRP RX, Roche Diagnostics GmbH, Mannheim, Germany). The method’s sensitivity was 0.3 mg/L.

The intra- and inter-assay coefficients of variation (CVs) were as follows: 1.1% and 1.6% for T-C, 1.2% and 2.5% for LDL-C, 1.8% and 2.2% for HDL-C, 1.1% and 2.0% for TG, 1.0% and 1.3% for glucose, and 1.5% and 4.9% for insulin.

Plasma IGF-1 levels were determined using an enzyme-labelled chemiluminescent immunometric assay (Mediagnost GmbH, Tuebingen, Germany). The sensitivity value was 10 ng/mL. Intra- and inter-assay CVs were 3.5% and 7%, respectively.

For each patient, the homeostatic model assessment of insulin resistance (HOMA-IR) was calculated using the formula (insulin [μIU/mL] × glucose [mmol/L])/22.5 [25].

### 2.4. Blood Pressure

Blood pressure was measured on the right arm using a sphygmomanometer with an appropriate cuff size for individuals with obesity. The subject was in a seated position and relaxed. The procedure was repeated three times at 10 min intervals between each measurement; the means of the three values for SBP and DBP were recorded.

### 2.5. Calculation of Framingham Risk Score and Vascular Age

The 2008 Framingham risk score (FRS) assessment was employed to determine the CVD risk and, additionally, the FRS-based vascular age (VA) [26].

The FRS algorithm considers age, T-C, HDL-C, SBP, ongoing hypertension treatment, smoking, and diabetes status and provides sex-specific results.

VA is defined as the age of a person with the same predicted CV risk but with all other risk factor levels in the normal ranges.

### 2.6. Determination of the Age Acceleration

Subjects provided a blood sample, which was collected in EDTA-containing tubes and immediately stored at −80 °C until assayed for epigenetic age.

Blood samples were thawed, and genomic DNA was extracted using the Wizard Genomic DNA Purification Kit (Promega, Madison, WI, USA) according to the manufacturer’s instructions.

Epigenetic age was calculated considering the methylation pattern of 5 CpG sites at five genes (ELOVL2, C1orf132/MIR29B2C, FHL2, KLF14, TRIM59) as reported elsewhere (18). The DNA samples (500 ng) were plated at a concentration of 25 ng/μL in plates of 96 wells each and were treated with sodium bisulphite using the EZ-96 DNA Methylation-Gold Kit (Zymo Research; Irvine, CA, USA) following the manufacturer’s instructions and eluted in 200 μL. Then, 10 μL of bisulphite-treated template DNA was added to 25 μL of GoTaq Hot Start Green Master mix (Promega, Madison, WI, USA), 1 μL of the forward primer (10 μM), and 1 μL of the 50 end-biotinylated reverse primer (10 μM) to set up a 50 μL PCR reaction. PCR cycling conditions and primer sequences have been previously reported (18).

The epigenetic age (YY) was calculated as follows:Y = 3.26847784751817 + 0.465445549010653 × methC7-ELOVL2 − 0.355450171437202 × methC1-C1orf132 + 0.306488541137007 × methC7-TRIM59 + 0.832684435238792 × methC1-KLF14 + 0.237081243617191 × methC2-FHL2

In this equation:−3.268477847518173.26847784751817 is the Intercept of the model, representing baseline epigenetic age;−0.4654455490106530.465445549010653 is the coefficient for methylation level at C7-ELOVL2C7-ELOVL2, derived from regression analysis [18].

Methylation levels were measured for each individual sample (*n* = 36 or *n* = 24, as indicated). These levels represent the percentage of methylated cytosines at the specified CpG sites and were derived from pyrosequencing measures. Coefficients reflect the contribution of each site to the estimated epigenetic age.

### 2.7. Statistical Analysis

Continuous variables, including values of demographic, lifestyle, biochemical, and clinical characteristics, and CVD outcomes in PWS vs. EOB groups, expressed as median [25th and 75th percentiles], were compared using the Wilcoxon signed-rank test. Categorical data were compared between the same groups with the Chi-squared test.

We calculated the correlations of chronological age with epigenetic age or VA by using Spearman’s rank correlation coefficient.

Age acceleration and vascular ageing were estimated by calculating the difference between epigenetic age or VA, respectively, and chronological age.

A series of Spearman’s rank correlations were performed to evaluate the associations of epigenetic age or age acceleration with the demographic, lifestyle, biochemical, and clinical characteristics measured at baseline.

The statistical analyses were performed using Sigma Stat v. 4.0 (SYSTAT Software, San Jose, CA, USA), while Prism GraphPad v. 10.2 (GraphPad Software, Boston, MA, USA) was used for graphic plotting.

*p*-values below 0.05 were considered statistically significant.

## 3. Results

Table 1 reports demographic, biochemical, and clinical characteristics of individuals with EOB and PWS, including the corresponding comparisons. Briefly, BMI, FFM (kg), FM (kg), SBP, insulin, HOMA-IR, and TG were significantly higher in the EOB than PWS group. Values of HbA1c were significantly lower in subjects with EOB than in individuals with PWS.

In the PWS group, epigenetic age was significantly younger than chronological age (*p* < 0.001). No individual with PWS had an epigenetic age older than the corresponding chronological age. In subjects with PWS, epigenetic age was significantly correlated with chronological age (r = 0.89; *p* < 0.0001) (Figure 1).

In the EOB group, no significant difference was present between chronological and epigenetic ages, although the two parameters were significantly correlated (r = 0.87; *p* < 0.0001) (Figure 1).

Although chronological age was similar between subjects with PWS and EOB, epigenetic age was significantly younger in the former than in the latter (Figure 2). Moreover, age acceleration was significantly lower in the PWS than in the EOB group (Figure 3).

Carrying out a series of correlations with demographic and biochemical parameters and clinical characteristics, taking into account only the significant results, in the EOB group, epigenetic age was positively associated with FM (%) (r = 0.40; *p* = 0.018) and HDL (r = 0.34; *p* = 0.040), and negatively with FFM (%) (r = −0.40; *p* = 0.019), while age acceleration was positively associated with SBP (r = 0.40; *p* = 0.017).

Repeating the same statistical analysis for the PWS group, there were no significant associations of epigenetic age or age acceleration with any demographic, biochemical, or clinical characteristics, including hormone replacement therapy with rhGH, duration of the treatment, and plasma levels of IGF-1.

No significant differences in VA and vascular ageing were detected between the PWS and EOB groups (Figure 2 and Figure 3). Nevertheless, within each group, VA was significantly associated with epigenetic age (Figure 1).

## 4. Discussion

The main finding of the present case–control study, carried out in sex- and age-matched adults with PWS and EOB, was the younger epigenetic age and lower age acceleration in the former than in the latter group.

We can suggest several explanations for these surprising findings, which contrast with the well-known increased CVD risk in obesity, a risk that, in the present study, was similar in both groups when estimated in terms of VA and vascular ageing [20,23]. At least hypothetically, in parallel with the finding of a younger epigenetic age, we would have expected a younger VA in individuals with PWS than in subjects with EOB. In fact, several clinical studies have demonstrated a direct correlation between vascular ageing and epigenetic age [21].

First of all, we have ruled out the impact of hormone replacement therapy with rhGH. Reportedly, GH deficiency (hyposomatotropism) is associated with long life expectancy (e.g., Laron’s syndrome), while GH overproduction (hypersomatotropism) has a higher mortality rate (e.g., acromegalia) [27]. In the present study, no associations with epigenetic age or age acceleration were found when considering individuals with PWS with GH deficiency, duration of rhGH treatment, or plasma levels of IGF-1, a biomarker of somatotropic function. This might mean that our subjects with PWS, particularly the GH-deficient ones, were adequately substituted.

Secondly, to explain the younger epigenetic age in individuals with PWS, we may not propose a different chronic inflammatory state, which has been reported to accelerate epigenetic ageing, for example, due to oxidative stress or mitochondrial dysfunction [28]. In fact, the subjects with PWS and EOB enrolled in the present study exhibited similar plasma levels of hsCRP, which is undoubtedly a “gross” marker of systemic inflammation [29].

Furthermore, many CVD risk factors, such as BMI, FM (kg), SBP, insulin resistance (HOMA-IR), and dyslipidaemia (TG), were higher in subjects with EOB than in individuals with PWS. The limited relevance of associations of epigenetic age or age acceleration with these CVD risk factors within either the PWS or EOB groups and the missing statistical significance of VA and vascular ageing between the two groups force us to exclude CVD outcomes as a causative determinant of the younger epigenetic age in individuals with PWS. Sustaining this argumentation does not mean denying any relationship between epigenetic age and vascular ageing [22]. In fact, as reported in the results of the present study, epigenetic age was positively associated with VA within each group, i.e., individuals with PWS and those with EOB. The (apparent) “epigenetic youth” depends on other causes.

PWS is a genetic disorder mapping to the imprinted 15q11.2-q13.3 locus, specifically at the paternally expressed SNORD116 region, including small nucleolar RNAs and noncoding host gene transcripts. In particular, SNORD116 is processed into several noncoding components and is hypothesised to orchestrate diurnal changes in neurodevelopment, feeding, and metabolism through epigenetic mechanisms, encompassing genes for methyl-binding proteins (mecp2), DNA:RNA hybrids binding proteins (setx), DNA demethylases (tet1, tet2, tet3), histone deacetylases (hdac3, hdac4, hdac5), DNA methyltransferases (dnmt1, dnmt3a), clock proteins (per2, per3, arntl), kinases involved in regulating cellular energy homeostasis (mtor), transcriptional regulator of E-box motif containing specific genes (neurod1) [30]. Based on this modern (and also fascinating) view of PWS as an epigenetic rather than simply a genetic disease [31], the younger epigenetic age that we have demonstrated for the first time in PWS is presumably a consequence of the different biological–molecular machinery underlying epigenetic modifications in these patients, including DNA methylation. Therefore, evaluating epigenetic age in PWS (as in the present study) might be misleading or inappropriate. More precisely, epigenetic age in PWS (and other syndromic diseases) might represent the outcome of an interaction of two distinct phenomena: epigenetic ageing in a context of altered epigenetics.

On the other hand, epigenetic age in PWS might be useful for demonstrating the effectiveness of future gene therapies, which have been successfully tested in animal models by restoring the function of the SNORD116 region [5].

Before closing, some limitations of our study should be mentioned.

First of all, the so-called “social” determinants, which, in the context of chronic disease, might have promoted psychobiological stress, which has been related to epigenetic alterations, including epigenetic ageing, have not been taken into account [32]. In this regard, recently, we have demonstrated that individuals with PWS, when compared to an age- and sex-matched healthy group, exhibit impairments in various aspects of quality of life and psychological well-being, including physical, behavioural, and social domains; surprisingly, higher vitality scores were observed in subjects with PWS, suggesting a preserved dimension of their psychological well-being [33]. Unfortunately, we administered no specific psychological questionnaire to the recruited subjects in this study. So, we cannot establish any causality relationship (or simply any association) between social determinants and epigenetics. Further studies are mandatory to explore this fascinating topic.

Second, a longitudinal study design would have provided more profound insights into how epigenetic age changes over time in individuals with PWS vs. those with EOB, which would be different from a cross-sectional one.

Third, we acknowledge the need to adjust our results for confounding factors such as physical activity, diet, medication use, sleep quality, etc. Unfortunately, though we are aware of the biological effects of these factors in epigenetic terms, we have not collected these data or possess only incomplete data. For each confounding factor (e.g., quality of sleep), validated (psychiatric/psychological) questionnaires and sophisticated (neurological) exams would have been administered, going beyond the purposes of our study.

## 5. Conclusions

Adults with PWS, when compared with sex- and age-matched individuals with EOB, exhibit a younger epigenetic age and a lower age acceleration. In contrast, VA and vascular ageing are similar between the two groups. Based on the results of the present study, the hypothesis that the clinical phenotype of PWS, including the high mortality rate, particularly in adulthood, may be due to accelerated epigenetic ageing should be rejected. Due to the (presumptive) SNORD116-dependent epigenetic regulation, an epigenetics-based methodology for evaluating ageing might be misleading when applied to individuals with PWS, characterised by a missing paternally imprinted SNORD116 gene region. Further clinical studies with different approaches (e.g., proteomics, metabolomics, glycomics) are mandatory to solve this issue [34].

## Figures and Tables

**Figure 1 jcm-14-01470-f001:**
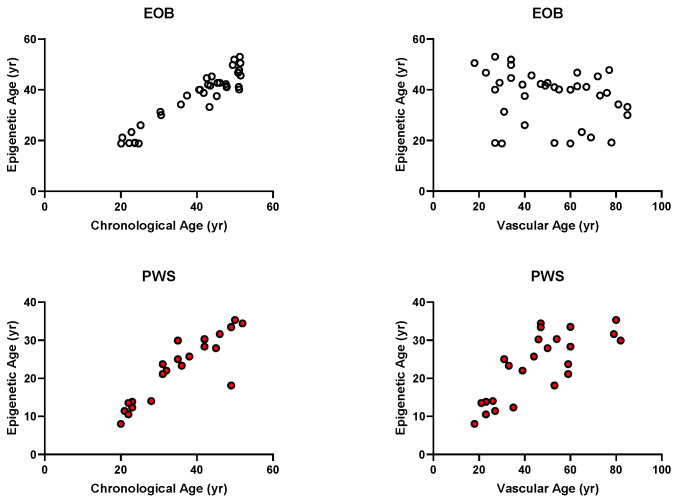
Correlations of epigenetic age with chronological age (**left panels**) or vascular age (**right panels**) for subjects with EOB (**top panels—white dots**) and PWS (**bottom panels—red dots**).

**Figure 2 jcm-14-01470-f002:**
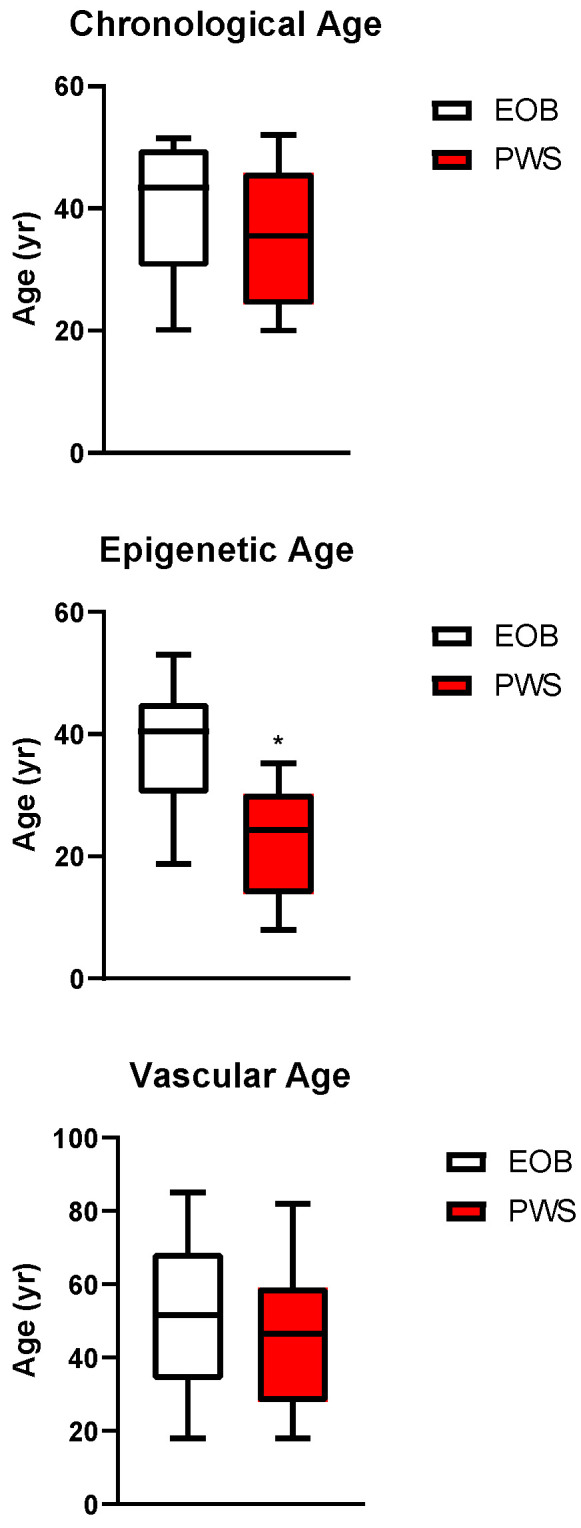
Chronological (top panel), epigenetic (middle panel), and vascular (bottom panel) ages in subjects with PWS compared to individuals with EOB. *: *p* < 0.05 vs. EOB.

**Figure 3 jcm-14-01470-f003:**
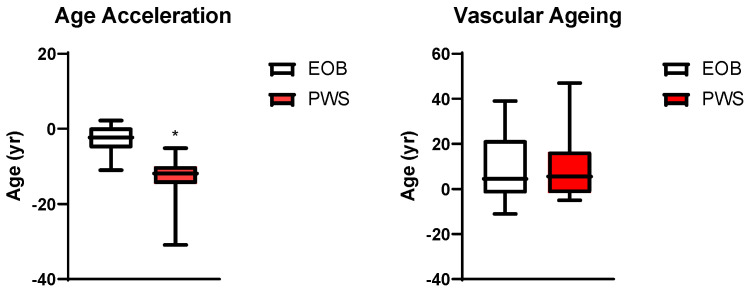
Age acceleration (left panel) and vascular ageing (right panel). *: *p* < 0.05: vs. EOB.

**Table 1 jcm-14-01470-t001:** The demographic, biochemical and clinical characteristics of the study population, subdivided into the two groups (EOB and PWS).

Parameter	EOB	PWS	*p*
N.	36	24	-
Sex (F/M)	19/17	11/13	ns
Age (year)	43.4 [30.6; 49.5]	36.8 [26.6; 45.3]	ns
Smoker (yes/no)	9/27	21/3	ns
BMI (kg/m^2^)	44.8 [41.2; 51.7]	36.8 [33.9; 44.8]	<0.001
WC (cm)	127.5 [113.0; 134.0]	113.5 [109.0; 122.8]	ns
FFM (kg)	55.2 [51.2; 74.8]	46.4 [43.3; 49.8]	<0.001
FFM (%)	47.4 [44.0; 51.4]	53.5 [45.3; 57.5]	ns
FM (kg)	67.3 [57.5; 77.3]	39.9 [35.4; 54.9]	<0.001
FM (%)	52.6 [48.6; 56.1]	46.5 [42.5; 54.7]	ns
SBP (mmHg)	140.0 [130.0; 145.0]	130.0 [120.0; 130.0]	=0.010
DBP (mmHg)	80.0 [80.0; 90.0]	80.0 [80.0; 82.5]	ns
HR (bpm)	83.0 [77.5; 89.3]	66.5 [64.0; 79.3]	<0.001
Antihypertensive Drugs (yes/no)	15/21	11/13	ns
Glucose (mg/dL)	100.0 [95.8; 108.0]	101.0 [85.3; 131.5]	ns
Insulin (mU/L)	24.9 [17.8; 35.7]	13.2 [8.1; 17.7]	<0.001
HOMA-IR	6.4 [4.1; 10.6]	3.5 [2.3; 5.3]	=0.002
HbA1c (%)	5.7 [5.5; 5.8]	6.0 [5.6; 7.1]	=0.025
DM (yes/no)	6/30	10/14	ns
Antidiabetic Drugs (yes/no)	7/29	11/13	ns
TC (mg/dL)	183.5 [167.0; 208.0]	191.0 [160.8; 214.0]	ns
HDL (mg/dL)	47.5 [39.8; 54.8]	50.5 [43.8; 54.8]	ns
LDL (mg/dL)	123.0 [104.5; 145.8]	122.0 [102.0; 147.0]	ns
TG (mg/dL)	138.0 [104.3; 180.5]	104.0 [74.0; 137.3]	=0.033
hsCRP (mg/dL)	0.7 [0.4; 1.2]	0.6 [0.3; 1.0]	ns
Epigenetic Age (year)	40.5 [30.9; 44.8]	24.4 [14.0; 30.2]	<0.001
VA (year)	51.5 [34.0; 67.5]	46.5 [30.0; 59.0]	ns
FRS (%)	4.5 [1.9; 15.0]	3.7 [1.4; 6.3]	ns

## Data Availability

The datasets used and/or analysed in the present study will be uploaded to www.zenodo.org and made available by the corresponding author upon a reasonable request.

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
