# Peer review of "Epigenetic Age in Prader–Willi Syndrome and Essential Obesity: A Comparison with Chronological and Vascular Ages"

_jcm, 2025, doi:10.3390/jcm14051470_

Round 1
Reviewer 1 Report
Comments and Suggestions for Authors
This study investigated epigenetic age, age acceleration, vascular age (VA), and vascular aging in PWS adults compared with a sex- and age-matched group of subjects with essential obesity. It found that PWS group presented a younger epigenetic age as well as a lower age acceleration when compared to EOB. No differences were found between VA and vascular aging in the two groups. This provides insights to the epigenetic aging and vascular aging in PWS adults. Overall, the study is presented in a clear way. However, authors will benefit from addressing the following points:
Major points:
1. No data are shown to support the conclusion written in line 268 to 271.
2. No data are shown to support the conclusion written in line 273 to 276.
3. The exact calculation or formula used to compute age acceleration and vascular aging is not articulated.
4. The conclusion at line 110 and 111 is confusing. PWS group has accelerated aging or less extent of aging?
5. Through the whole manuscript, data from PWS group were not separated based on their GH treatment regime. It is unclear to the readers how the conclusion at line 294, “In the present study, no associations with epigenetic age or age acceleration were found when considering PWS subjects with GH deficiency, duration of rhGH treatment, or plasma levels of IGF-1, a biomarker of somatotropic function.”, was drawn.
Minor points:
1. At line 278, it was written Fig.s 2-3, which does not exist in the manuscript.
2. What is hsPCR (mg/dL)?
Reviewer 2 Report
Comments and Suggestions for Authors
The paper is described remarkably well, particularly in the methodology section, in which the detailed level of laboratory techniques is a highlight and reveals the author's care in producing the results. The statistics are well-designed, and the study will certainly provide new perspectives for the management of Prader-Willi syndrome (PWS).
However, the text presents the context of obesity in a totally biological way, and disregards the social determinants, which are so important in the evaluation of a chronic disease. This may be a limitation of the study, but in the context of this well described text, I suggest that the discussion be supplemented with reflections on the social determinants.
Reviewer 3 Report
Comments and Suggestions for Authors
Manuscript Title: Epigenetic Age in Prader-Willi Syndrome and Essential Obesity: A Comparison with Chronological and Vascular Ages
Journal: Journal of Clinical Medicine (JCM)
Summary of the Study
This manuscript investigates epigenetic age, age acceleration, vascular age (VA), and vascular aging in adults with Prader-Willi Syndrome (PWS) compared to individuals with essential obesity (EOB). The study hypothesizes that PWS patients experience accelerated biological aging, which may explain their higher mortality rate in adulthood. However, contrary to expectations, the findings reveal that PWS individuals have a younger epigenetic age and lower age acceleration than their EOB counterparts, despite both groups exhibiting similar vascular aging.
The authors argue that these results challenge the hypothesis of accelerated biological aging in PWS and suggest that epigenetic-based aging assessments might be misleading in this condition due to SNORD116-dependent epigenetic dysregulation.
Strengths of the Study
Innovative Focus on Epigenetic Aging in PWS:
This is one of the first studies assessing epigenetic age in PWS beyond telomere length analysis, which adds a novel dimension to understanding aging in this condition.
Well-Defined Research Question and Justification:
The study presents a clear and logical research question, addressing an important gap in the understanding of aging in PWS.
Robust Case-Control Design:
The sex- and age-matched case-control design strengthens the validity of comparisons between PWS and EOB groups.
Comprehensive Methodological Approach:
The use of DNA methylation-based epigenetic clocks for estimating biological age is an appropriate and contemporary method.
Statistical Rigor:
The manuscript employs relevant statistical tests, including Wilcoxon signed-rank tests, Spearman’s rank correlations, and regression analyses to evaluate associations.
Clinical Relevance:
Findings could have implications for understanding aging in PWS and may influence future therapeutic strategies, including potential gene therapy interventions.
Limitations and Areas for Improvement
Sample Size Concerns:
The study is limited by its small sample size (PWS: n=24, EOB: n=36), which may reduce statistical power and generalizability. A larger cohort would strengthen conclusions.
Lack of Longitudinal Data:
Since aging is a progressive process, a cross-sectional design limits the ability to infer causality or detect long-term trends in epigenetic aging.
Potential Confounding Factors Not Fully Addressed:
Factors such as medication use, diet, physical activity levels, and sleep quality, which are known to influence epigenetic aging, are not comprehensively controlled for or discussed.
Reliance on a Single Epigenetic Clock Algorithm:
Only one epigenetic age estimation model is used. Given the variability among different epigenetic clocks (e.g., Horvath’s, Hannum’s, GrimAge), results might be different with alternative methodologies.
Interpretational Challenges of Epigenetic Data:
The authors acknowledge that epigenetic aging measures might be misleading in PWS due to SNORD116-related dysregulation. However, further clarification on how this could impact the study’s conclusions would be beneficial.
No Direct Functional Validation of Epigenetic Mechanisms:
While the study discusses potential epigenetic alterations in PWS, no functional validation (e.g., gene expression analysis, chromatin modifications) is provided to support the findings.
Discussion Could Be More Balanced:
The authors should consider alternative explanations for their findings beyond epigenetic dysregulation in PWS, such as metabolic adaptations or differential inflammatory responses.
Limited Exploration of Clinical Implications:
While the study suggests that epigenetic aging metrics may not be suitable for assessing biological age in PWS, there is minimal discussion on how this finding could impact clinical management or treatment strategies.
Recommendations for Improvement
Increase Sample Size:
If feasible, recruiting a larger cohort would enhance statistical power and external validity.
Longitudinal Analysis:
A longitudinal study design would provide deeper insights into how epigenetic age changes over time in PWS versus EOB.
Use Multiple Epigenetic Age Clocks:
Employing different epigenetic age predictors (e.g., GrimAge, PhenoAge) could strengthen the reliability of findings.
Adjust for Additional Confounders:
Consider controlling for variables such as physical activity, diet, and medication use to refine the analysis.
Functional Follow-Up Studies:
Future research could include transcriptomic or proteomic analyses to validate potential biological mechanisms underlying epigenetic changes in PWS.
Expand the Discussion Section:
Provide a more balanced discussion, incorporating alternative explanations and exploring the broader implications for aging research and clinical practice.
Clarify the Clinical Relevance of Findings:
The manuscript should elaborate on how these results could influence future diagnostic or therapeutic approaches in PWS.
Final Verdict
The manuscript presents a well-designed and methodologically sound study with novel insights into epigenetic aging in PWS. The findings are intriguing and challenge the notion of accelerated biological aging in this population. However, limitations related to sample size, cross-sectional design, confounding factors, and interpretational challenges of epigenetic data should be addressed to strengthen the study’s conclusions.
Round 2
Reviewer 1 Report
Comments and Suggestions for Authors
Thanks for addressing the questions, but it will great if authors could further answer my following questions.
Q1-Q2
If no data in the manuscript are used to support those conclusions, pls revise the text to match it.
